# Antioxidant and Biological Activities of Mahajanaka Mango Pulp Extract in Murine Models

Narisara Paradee [1], Rattanaporn Janthip [2], Tawat Taesothikul [2], Duangta Kanjanapothi [2], Kornvipa Settakorn [1], Somdet Srichairatanakool [1] and Pimpisid Koonyosying [1,*]

[1] Department of Biochemistry, Faculty of Medicine, Chiang Mai University, Chiang Mai 50200, Thailand; narisara.p@cmu.ac.th (N.P.); stkornvipa@gmail.com (K.S.); somdet.s@cmu.ac.th (S.S.)

[2] Faculty of Pharmacy, Payap University, Chiang Mai 50000, Thailand; rjanthip@gmail.com (R.J.); ttaesoti@mail.med.cmu.ac.th (T.T.); drkanjanapothi@hotmail.com (D.K.)

[*] Correspondence: pimpisid.k@cmu.ac.th; Tel.: +66-5393-5322

**Abstract:** Mahajanaka mango, a hybrid cultivar of *Mangifera indica* Linn., is a highly nutritional fruit that is popularly consumed in Thailand. It has been used in traditional medicine due to its abundance of phytonutrients. The present study aimed to investigate the chemical compositions and antioxidant activity of Mahajanaka mango pulp extract (MPE) in vitro. Additionally, we examined its biological activities, including the analgesic, anti-inflammatory, gastroprotective, and hepatoprotective effects of MPE, in murine models. MPE exhibited high levels of phenolic compounds, mangiferin, β-carotene, and vitamin C, and it potentially showed antioxidant properties in an ABTS scavenging assay. The animal results have revealed that oral administration of MPE (1000 mg/kg body weight (BW)) significantly decreased acetic acid-induced writhing responses in mice. Interestingly, local applications of MPE at 1 mg/ear ameliorated ethyl phenylpropiolate (EPP)-induced ear edema, while gavage of MPE at 1000 mg/kg BW significantly decreased carrageenan-induced hind paw edema in rats. MPE can also protect against gastric ulcers induced by stress, hydrochloric acid/ethanol, and indomethacin in rats. Indeed, MPE (250 mg/kg BW) markedly lowered the level of serum alanine aminotransferase activity and hepatic lipid accumulation in rats with $CCl_4$- and paracetamol-induced hepatotoxicity. Taken together, the findings suggested that MPE exerts potent antioxidant, analgesic, anti-inflammatory, gastroprotective, and hepatoprotective effects.

**Keywords:** *Mangifera indica*; Mahajanaka mango; antioxidant; analgesic; anti-inflammatory; gastroprotective; hepatoprotective

## 1. Introduction

Mango (*Mangifera indica* Linn) not only is a popular edible fruit but also is widely regarded for its medicinal properties [1,2]. Mahajanaka mango is one of the most popular cultivars in Thailand and is a hybrid cultivar that has been developed by crossing the Nang Klang Wan and Sunset cultivars. This mango is recognized for its attractive exocarp color which ranges from red to yellowish-red when the fruit is ripe. It is also known for its unique taste, aroma, and high nutritional value. Importantly, it contains bioactive compounds such as mangiferin, gallotannins, gallic acid, quercetin, and catechins, all of which are known to be beneficial to health. Furthermore, it possesses a range of antioxidant, anti-microbial, and anti-inflammatory properties [3–6]. Surprisingly, mangiferin has protective effects against gastric inflammation in *Helicobacter pylori*-induced human gastric adenocarcinoma AGS cells [7]. It has also exhibited anti-inflammatory and antipyroptotic effects in lipopolysaccharide (LPS)-induced mouse bone-marrow-derived macrophages [8]. Moreover, mango pulp extracts can protect against thioacetamide-induced hepatotoxicity in male rats [9].

Inflammation is a natural defense mechanism that can be characterized by local redness, heat, swelling, pain, and a loss of tissue function, all of which are initiated by the

immune response to harmful stimuli such as pathogens, toxins, or certain tissue injuries [10]. It involves a complex cascade of events that results in the recruitment of immune cells, the release of proinflammatory cytokines, and the production of reactive oxygen species and other mediators [11,12]. However, excessive or prolonged inflammation can lead to tissue damage and has been implicated in the development of various diseases. These include peptic ulcers and a number of liver diseases such as steatosis and cirrhosis [13,14]. Nonsteroidal anti-inflammatory drugs (NSAIDs), such as aspirin, ibuprofen, and indomethacin, are commonly used to manage pain and inflammation, but their long-term use has been associated with adverse effects such as gastrointestinal complications, liver toxicity, renal dysfunction, and an increased risk of cardiovascular events [15,16]. As a result, there has been growing interest in the development of natural anti-inflammatory agents with fewer side effects.

Globally, peptic ulcers and liver diseases are major public health concerns. Therefore, the identification of effective alternative medications to treat these conditions will be crucial [17,18]. Accordingly, the pulp of the Mahajanaka mango exhibits nutraceutical properties that can be implemented in various therapeutic applications. In this regard, the present study aimed to evaluate the chemical compositions and antioxidant activity of Mahajanaka mango pulp extract (MPE) in vitro and biological activities (analgesic, anti-inflammatory, and gastroprotective and hepatoprotective activities) of MPE in animal models.

## 2. Materials and Methods

### 2.1. Chemicals and Reagents

Acetonitrile, ethanol, and methanol were purchased from Fisher Scientific Company (Hampton, NH, USA). Acetic acid, hydrochloric acid (HCl), and L-ascorbic acid were purchased from Merck Company (Darmstadt, Germany). Carbon tetrachloride, carrageenan, cimetidine, 2,6-dichlorophenolindophenol (DCPIP), dimethyl sulfoxide (DMSO), ethyl phenylpropiolate (EPP), phenylbutazone, and indomethacin were obtained from Sigma-Aldrich Chemicals Company (St. Louis, MO, USA). Aspirin, pentobarbital sodium, and paracetamol were purchased from a local drug store located in Maharaj Nakorn Chiang Mai Hospital, Faculty of Medicine, Chiang Mai University, Chiang Mai, Thailand.

### 2.2. Preparation of Mahajanaka Mango Pulp Extract (MPE)

Thai ripe mango (*Mangifera indica*) fruits, cultivar Mahajanaka, were harvested from an orchard located in Doi Lo District, Chiang Mai, Thailand, and botanically examined at the Herbarium, Faculty of Pharmacy, Chiang Mai University (voucher specimen number: 0023368). Firstly, the fruits were washed and peeled. Then, the pulp was chopped and extracted with DI water at a ratio of 1:1 (*w/v*) at room temperature for 24 h. Afterward, the water extract was filtrated through filter paper (Cellulose type, Whatman No. 1, Maidstone, UK), lyophilized to dryness using a freeze-dry lyophilizer, and stored at −20 °C for further experiments.

### 2.3. Determination of Chemical Compositions in MPE

2.3.1. Identification of Phenolic Compounds

Phenolic compounds in MPE were identified using the high-performance liquid chromatography/electrospray ionization mass spectrometry (HPLC/ESI-MS) method [19]. Analyses were performed on a Waters platform ZMD 4000 system, which comprised a Waters 2690 HPLC, a Waters 996 photodiode array detector, and a micro ZMD mass spectrometer equipped with an electrospray source. The source block temperature was set at 80 °C, the desolvation temperature at 150 °C, and the capillary voltage at 3 kV. Chromatography was carried out with a 10 µL sample injection onto a 250 mm × 4.6 mm C-18 column, using a mobile phase of water–methanol–acetic acid (83:15:2) with a flow rate of 1 mL/min for 60 min. The ESI source employed argon as a cone gas (50 L/h) and nitrogen as a desolvation gas (50 L/h), with a flow rate of 70 mL/min. The ESI-MS was configured to record in the range of $m/z$ 100–500, with a scan time of 1 s and an interscan delay of 0.1 s.

### 2.3.2. Determination of Mangiferin Content

Mangiferin content in MPE was determined using the high-performance liquid chromatography/diode array detector (HPLC/DAD) method, as has been previously established by Prabhu et al. [20]. In this analysis, MPE was reconstituted homogenously in methanol, and the sample (10 μL) was injected into the HPLC/DAD system (Agilent Technologies, Santa Clara, CA, USA), fractionated on a column (C18 type, 4.6 mm × 250 mm, 5 μm particle size, Agilent Technologies, Santa Clara, CA, USA), and eluted by a mobile-phase solvent composed of methanol–3.5% acetic acid (33:67) at a flow rate of 0.5 mL/min, and eluents were detected at a wavelength of 254 nm with a DAD.

### 2.3.3. Determination of β-Carotene Content

β-carotene content was determined as previously established by Tee and Lim [21]. In brief, mango pulp was saponified with 100% potassium hydroxide in ethanol at 85 °C for 30 min and then extracted with hexane. The extract was concentrated using rotary evaporators. The sample was reconstituted with methanol and injected into the HPLC/DAD system (Agilent Technologies, Santa Clara, CA, USA). Fractionation occurred on a C18 column (4.6 mm × 250 mm, 5 μm particle size, Agilent Technologies, Santa Clara, CA, USA), with elution by a mobile-phase solvent composed of acetonitrile–methanol–ethyl acetate (88:10:2) at a flow rate of 1 mL/min. Eluents were detected at a wavelength of 436 nm with a DAD. The results were expressed as mg β-carotene/100 g fresh weight.

### 2.3.4. Determination of Vitamin C Content

Vitamin C content in MPE was determined using the 2,6-dichlorophenolindophenol (DCPIP) titration method [22]. In this assay, MPE was mixed with 3% phosphoric acid in DI water and then titrated with DCPIP solution until a pink color appeared. The volume of DCPIP was recorded. Vitamin C concentration was calculated from an L-ascorbic acid standard.

### 2.3.5. Determination of Total Phenolic Content

Total phenolic content in MPE was determined using the Folin–Ciocalteu colorimetric method [23]. Briefly, extracts or gallic acid (standard) was incubated with 10% Folin–Ciocalteu reagent and 7% sodium carbonate at room temperature for 30 min, prior to the measurement of optical density (OD) at 765 nm. A standard curve was constructed by plotting different concentrations of gallic acid against OD. Total phenolic content was expressed as mg of gallic acid equivalent (GAE)/g extract.

### 2.4. Determination of Antioxidant Activity in MPE

The ABTS radical-scavenging activity assay was employed to determine the antioxidant activity [23]. The working ABTS radical (ABTS$^{\bullet+}$) solution was prepared by oxidizing a 7 mM ABTS stock solution with 2.45 mM $K_2S_2O_8$ (1:1, *v/v*) and allowed to undergo storage at room temperature for 12–16 h. Prior to the assay, the working ABTS$^{\bullet+}$ solution was diluted with DI water, and its absorbance was measured at 734 nm until it reached an OD of $0.70 \pm 0.02$. For the assay, extracts or Trolox (standard) was mixed with the diluted ABTS$^{\bullet+}$ solution in the dark for 6 min. The absorbance values were then measured at 734 nm. The result was reported as mg Trolox equivalent antioxidant capacity (TEAC)/g extract.

### 2.5. Animal Care

The National Laboratory Animal Center, Mahidol University Salaya Campus, Nakorn Pathom, Thailand, provided Swiss albino mice and Wistar rats. The animals were kept in separate cages with free access to commercial food (No. C.P. 082, Perfect Companion Group Company Limited, Bangkok, Thailand) and water. The animals were housed with a 12 h light/dark cycle at room temperature (20–22 °C) at a humidity of $50 \pm 10\%$. Before the experiments, all animals were acclimatized to laboratory conditions for one week.

## 2.6. Ethics

All procedures for the animal experiments were approved by the Animal Ethical Committee from the Faculty of Medicine, Chiang Mai University, Thailand (Protocol Number 21/2554).

## 2.7. Investigation of Analgesic Activity on Acetic Acid-Induced Writhing Responses in Mice

An injection of acetic acid was used to induce writhing in mice. Stretching the abdomen, extending the hindlimbs, or turning the trunk were all signs of writing (twisting). Acetic acid-induced writhing responses in mice were assessed according to a previously described method [24]. Twenty-five male mice, weighing 40–60 g, were randomly categorized into five groups with five mice in each group. Deionized water (DI) (5 mL/kg BW), aspirin (150 mg/kg BW), and MPE at doses of 250, 500, and 1000 mg/kg BW were orally administered to the mice in each group 30 min before stimulation of writhing. After 30 min, 0.75% ($v/v$) acetic acid (10 mL/kg) was intraperitoneally injected into all mice. After 5 min of the injection, incidences of writhing were counted for 15 min. Analgesic activities were then determined and expressed as the percentage of inhibition of the incidences of writhing when compared with rats that had not received acetic acid injections.

## 2.8. Investigation of Anti-Inflammatory Activity in Rats
### 2.8.1. Ethyl Phenylpropiolate-Induced Ear Edema

For the purposes of this investigation, EPP is an irritant that was applied to the rats' ears to induce acute inflammation and edema. Thus, EPP-induced ear edema in rats was assessed following the method of Sireeratawong et al. [25]. Thirty male rats, weighing 60–80 g, were randomly categorized into five groups (n = 6 each). Then, 20 µL of 5% ($v/v$) DMSO previously dissolved in acetone, phenylbutazone (1 mg/ear), and MPE (1, 2.5, and 5 mg/ear) were topically applied to the inner and outer surfaces (10 µL each side) of the rats' ears. After that, EPP (1 mg, 20 µL/ear) was topically applied to the rats' ears. The ear thickness was measured using a digital Vernier caliper at 15, 30, 60, and 120 min after induction of ear edema. Anti-inflammatory activities were expressed as the percentage of inhibition of ear edema when compared with the control (DMSO).

### 2.8.2. Carrageenan-Induced Hind Paw Edema Model

Carrageenan is a potent chemical that can initiate local inflammatory responses and edema. Carrageenan-induced hind paw edema in rats was assessed using the method established by Winter et al. [26]. Thirty male rats, weighing 100–120 g, were randomly categorized into five groups with six rats in each group. DI water (5 mL/kg BW), aspirin (300 mg/kg BW), and MPE (250, 500, and 1000 mg/kg BW) were orally administered to the rats in each group 1 h before induction of inflammation. Afterward, 1% ($v/v$) carrageenan (50 µL/paw) was intradermally injected into the right hind paw of each rat. The edema volume of the hind paw was then measured using a digital Vernier caliper at 1, 3, and 5 h after the carrageenan injections. Anti-inflammatory activities were expressed as the percentage of inhibition of hind paw edema when compared with the control.

## 2.9. Investigation of Gastroprotective Effects in Rats
### 2.9.1. Water Immersion and Restraint Stress (WIRS)-Induced Gastric Ulceration

Stress is one of the factors that can cause gastric ulcers. Thirty male rats, weighing 180–200 g, were randomly categorized into five groups (n = 6 each). Gastric ulceration was then induced according to a previously described method [24]. Rats were fasted for 48 h before receiving treatment but had free access to water except for 1 h prior to receiving the treatment. DI water (5 mL/kg BW), cimetidine (100 mg/kg BW), and MPE (250, 500, and 1000 mg/kg BW) were then orally administered to the rats in each group. After 1 h treatments, each rat was individually restrained in a cage and immersed up to its xiphoid in a temperature-controlled water bath (22 °C) for 5 h to induce peptic ulcers. Rats were anesthetized by intraperitoneal injection of pentobarbital sodium (40 mg/kg BW), and their

stomachs were dissected and rinsed with normal saline. The stomachs were opened along the great curvature and photographed. Under light microscopy, the lengths of the stomach lesions were measured and expressed as the total of the lesion lengths. Gastroprotective effects were expressed as the percentage of inhibition of the gastric ulcers when compared with the control.

### 2.9.2. Hydrochloric Acid/Ethanol-Induced Gastric Ulceration

The stomach mucosal barrier of rats was damaged by the administration of HCl and ethanol, resulting in inflammation and gastric ulcers. HCl/ethanol-induced gastric ulcers in rats were investigated following the method of Mizui and Doteuchi [27]. Thirty male rats, weighing 180–200 g, were randomly divided into five groups (n = 6 each). Rats were fasted for 48 h before receiving the treatment but had free access to water except for 1 h before initiation of the experiment. DI water (5 mL/kg BW), cimetidine (100 mg/kg BW), and MPE (250, 500, and 1000 mg/kg BW) were orally given to the rats in each group 1 h before induction of the gastric ulcers. The rats were then orally fed with HCl in ethanol (36.5% *v/v*, 5 mL/kg BW). After 1 h, the rats were anesthetized by intraperitoneal injection of pentobarbital sodium (40 mg/kg BW), and their stomachs were dissected and rinsed with normal saline. The stomachs were then opened and photographed. The total lengths of the gastric lesions were determined under light microscopy. Gastroprotective effects were expressed as the percentage of inhibition of the gastric ulcers when compared with the control.

### 2.9.3. Indomethacin-Induced Gastric Ulcer Model

Indomethacin is an NSAID that can cause gastric mucosal injuries and gastric ulcers. Gastric ulcers in rats were also induced by indomethacin according to the method established by Djahanguiri [28]. Similarly, thirty male rats, weighing 180–200 g, were randomly divided into five groups (n = 6 each). Rats were fasted for 48 h before treatments but had free access to water except for 1 h before initiation of the experiment. DI water (5 mL/kg BW), cimetidine (100 mg/kg BW), and MPE (250, 500, and 1000 mg/kg BW) were orally fed to the rats in each group 1 h before induction of the gastric ulcers. Indomethacin (30 mg/kg BW) was then subcutaneously (*sc*) injected into the abdominal cavities of the rats. After 5 h, the rats were anesthetized by *ip* injection of pentobarbital sodium (40 mg/kg BW), and their stomachs were dissected and rinsed with normal saline. The stomachs were then opened and photographed. The total lengths of the gastric lesions were determined under light microscopy. Gastroprotective effects were expressed as the percentage of inhibition of the gastric ulcers when compared with the control.

### *2.10. Evaluation of Hepatoprotective Effects in Rats*

2.10.1. Carbon Tetrachloride-Induced Hepatotoxicity

$CCl_4$-induced lipid peroxidation, as well as lipid accumulation in the liver, is known to lead to fatty livers. $CCl_4$-induced hepatotoxicity in rats was achieved according to a previously described method [29]. Thirty male rats, weighing 180–200 g, were randomly categorized into seven groups (n = 8 each). Rats in the control groups were orally fed with DI water (5 mL/kg BW) and MPE (1000 mg/kg BW), while rats in the $CCl_4$-induced hepatotoxicity group were orally fed with DI water (5 mL/kg BW), silymarin (100 mg/kg BW), and MPE (250, 500, and 1000 mg/kg BW) once a day for ten days. After 1 h treatments, rats in the $CCl_4$-induced hepatotoxicity group were injected with $CCl_4$ (2 mg/kg BW, *sc*) on days 9 and 10. On day 11, rats were anesthetized by injection of pentobarbital sodium (40 mg/kg BW, *ip*), and their blood was collected from the heart into serum-separating tubes. The blood was centrifuged at $3000 \times g$ at room temperature for 10 min to obtain the serum used for the determination of ALT and AST activity, as will be described below. Each liver was dissected, weighed, and preserved in 10% paraformaldehyde for histochemical analysis.

### 2.10.2. Paracetamol-Induced Hepatotoxicity

High-dose consumption of paracetamol can cause hepatotoxicity and acute liver failure. In our study, thirty male rats, weighing 180–200 g, were randomly divided into seven groups (n = 8 each) and hepatotoxicity was induced by oral administration of paracetamol with gavage according to a previously described method with slight modifications [30]. The rats were fed with DI water (5 mL/kg BW), silymarin (100 mg/kg BW), and MPE (250, 500, and 1000 mg/kg BW) once a day for ten days. On day 9, rats in the paracetamol-induced hepatotoxic groups were orally fed paracetamol (2.5 g/kg BW) at 1 h after treatment of the compounds. On day 11, all rats were anesthetized by *ip* injection of pentobarbital sodium (40 mg/kg BW). Blood samples were then collected from the heart into serum-separating tubes for biochemical analysis. The livers were dissected, weighed, and preserved in paraformaldehyde (10% *v*/*v*) for histopathological examination.

### 2.10.3. Colorimetric Determination of Liver Function Enzymes

Serum AST and ALT activities were determined using a Randox Daytona Chemistry Analyzer (Randox Laboratories Limited, County Antrim, UK) according to the manufacturer's instructions.

### 2.10.4. Histopathological Examination

Each liver was embedded in a paraffin block, cut into slides, deparaffinized, and stained with hematoxylin and eosin (H&E) dye. The liver tissue sections were examined in terms of their histopathological status and photographed under light microscopy by a qualified expert clinical pathologist at the Department of Pathology, Faculty of Medicine, Chiang Mai University, Thailand.

### *2.11. Statistical Analysis*

The results are represented as mean values $\pm$ standard deviations (SDs). A one-way analysis of variance (ANOVA) followed by post hoc Duncan tests was conducted to assess statistical significance, considering results with $p < 0.05$ as indicative of statistical significance.

## 3. Results

### *3.1. Characterization of Phenolic Compounds*

Based on the findings from the analysis using high-performance liquid chromatography/electrospray ionization mass spectrometry (HPLC/ESI-MS), as depicted in Figure 1 and Table 1, MPE comprised identifiable constituents such as quercetin-*O*-glucuronide, quercetin-*O*-rutinoside, kaempferol-*O*-rutinoside, quercetin-*O*-glucoside, quercetin, and kaempferol. These findings indicate that the MPE contained quercetin, kaempferol, and their respective derivatives.

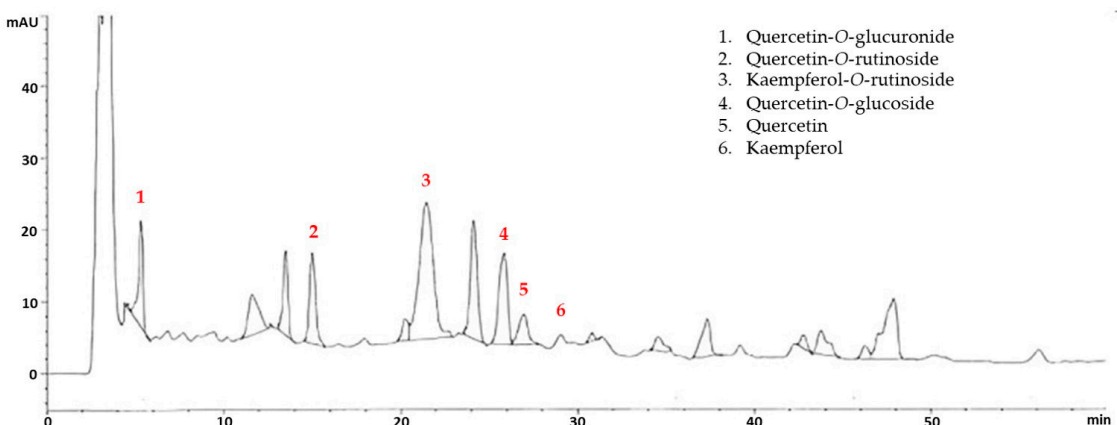

**Figure 1.** HPLC/ESI-MS analysis of phenolic compounds in MPE.

**Table 1.** HPLC/ESI-MS assignments of phenolic compounds detected in MPE.

| $R_T$ (min) | Assigned Compound | *m/z* |
|---|---|---|
| 5.327 | Quercetin-*O*-glucuronide | 478.1 |
| 15.008 | Quercetin-*O*-rutinoside | 610.2 |
| 21.491 | Kaempferol-*O*-rutinoside | 597.0 |
| 25.861 | Quercetin-*O*-glucoside | 465.0 |
| 26.932 | Quercetin | 300.1 |
| 29.023 | Kaempferol | 283.3 |

### 3.2. Chemical Composition and Antioxidant Measurement of MPE

According to our chemical composition analysis (Table 2), the MPE exhibited high levels of mangiferin (210 $\pm$ 13 µg/g extract), β-carotene (5.8 $\pm$ 0.1 mg/g fresh weight), and vitamin C (690 $\pm$ 30 µg/g extract). The MPE also contained a total phenolic content of 85.6 $\pm$ 7.7 mg gallic acid equivalent (GAE)/g extract. Moreover, the MPE demonstrated scavenging activity against ABTS radicals, with 192.4 $\pm$ 3.2 mg Trolox-equivalent antioxidant capacity (TEAC)/g extract. These findings emphasize that the MPE is rich in mangiferin, β-carotene, and vitamin C and possesses potent antioxidant properties.

**Table 2.** Mangiferin, β-carotene, vitamin C, total phenolic content antioxidant activity of MPE. Data are expressed as mean $\pm$ SD values (n = 3).

| Chemical Composition | MPE |
|---|---|
| Mangiferin (µg/g extract) | 210 $\pm$ 13 |
| β-Carotene (mg/100 g fresh weight) | 5.8 $\pm$ 0.1 |
| Vitamin C (µg/g extract) | 690 $\pm$ 30 |
| Total phenolic content (mg GAE/g extract) | 85.6 $\pm$ 7.7 |
| ABTS radical-scavenging activity (mg TEAC/g extract) | 192.4 $\pm$ 3.2 |

### 3.3. Analgesic Effects of MPE on Acetic Acid-Induced Writhing Responses in Mice

MPE was dosed into 250, 500, and 1000 mg/kg body weight (BW) and then orally administered to mice by gavage (*po*). In our study, intraperitoneal (*ip*) injections of acetic acid into mice resulted in approximately 25 incidences of abdominal writhing (Table 3). Aspirin, which comprises acetylsalicylic acid, significantly reduced the frequency of acetic acid-induced writhing and resulted in 75% inhibition of writhing when compared with the DI water control group. The consumption of MPE (250, 500, and 1000 mg/kg BW) decreased incidences of writhing in a concentration-dependent manner, producing 43, 55, and 65% inhibitions in writhing, respectively, wherein only a dose of 1000 mg/kg BW demonstrated a significant effect. The results imply that MPE produced an analgesic effect through the prevention of acetic acid-induced writhing in mice.

**Table 3.** Analgesic effects of MPE on acetic acid-induced writhing responses in mice. Data are expressed as mean $\pm$ SD values (n = 5). * $p < 0.05$ when compared with DI water.

| Group | Incidences of Writhing | % Inhibition |
|---|---|---|
| DI water (5 mL/kg BW) | 24.60 $\pm$ 1.50 | - |
| Aspirin (150 mg/kg BW) | 6.20 $\pm$ 1.66 * | 75 |
| MPE (250 mg/kg BW) | 14.00 $\pm$ 3.58 | 43 |
| MPE (500 mg/kg BW) | 11.20 $\pm$ 5.03 | 55 |
| MPE (1000 mg/kg BW) | 8.60 $\pm$ 2.89 * | 65 |

Abbreviations: DI = deionized, MPE = Mahajanaka mango pulp extract.

### 3.4. Anti-Inflammatory Effects of MPE

#### 3.4.1. Ethyl Phenylpropiolate (EPP)-Induced Ear Edema in Rats

EPP was topically applied to the ears of rats to induce ear edema, the thickest of which was measured at 60 min. The treated areas were then maintained for 120 min

(Table 4). Phenylbutazone significantly suppressed the ear swelling at 15, 30, 60, and 120 min, indicating 80, 75, 49, and 49% inhibition of ear swelling, respectively, when compared with the DI water control group. At each time point, the MPE at doses of 1, 2.5, and 5 mg/ear significantly reduced the ear edema thickness when compared with the DI water control group. Interestingly, the MPE at a dose of 1 mg/ear resulted in 74, 50, 32, and 47% inhibition of ear swelling at 15, 30, 60, and 120 min, respectively, which was found to be more effective than MPE at doses of 2.5 and 5 mg/ear. The results indicate that the MPE displayed anti-inflammatory activity through the limiting of EPP-induced ear edema in rats.

**Table 4.** Anti-inflammatory activity of MPE on EPP-induced ear edema in rats (n = 6). Data are expressed as mean $\pm$ SD values. Accordingly, * $p < 0.05$ when compared with DI water.

| Group | Measure | Time (Min) | | | |
|---|---|---|---|---|---|
| | | **15** | **30** | **60** | **120** |
| DI water (1 mL/ear) | Thickness (μm) | 195.0 ± 8.9 | 277.2 ± 7.0 | 356.1 ± 14.7 | 335.6 ± 11.3 |
| Phenylbutazone (1 mg/ear) | Thickness (μm) Inhibition (%) | 40.0 ± 11.2 * 80 | 69.5 ± 8.5 * 75 | 182.8 ± 7.2 * 49 | 171.1 ± 10.9 * 49 |
| MPE (1 mg/ear) | Thickness (μm) Inhibition (%) | 50.6 ± 3.4 * 74 | 138.9 ± 6.4 * 50 | 243.9 ± 5.5 * 32 | 178.3 ± 9.4 * 47 |
| MPE (2.5 mg/ear) | Thickness (μm) Inhibition (%) | 145.6 ± 8.6 * 25 | 202.8 ± 8.5 * 27 | 293.3 ± 10.1 * 18 | 260.0 ± 14.2 * 23 |
| MPE (5 mg/ear) | Thickness (μm) Inhibition (%) | 165.0 ± 9.4 * 15 | 214.4 ± 13.4 * 23 | 293.9 ± 8.8 * 18 | 268.9 ± 14.5 * 20 |

Abbreviations: DI = deionized, MPE = Mahajanaka mango pulp extract.

### 3.4.2. Carrageenan-Induced Hind Paw Edema in Rats

Carrageenan was administered to the hind paws of rats to induce paw swelling, the largest of which was acknowledged at 5 h (Table 5). A reference dosage of the NSAID aspirin significantly decreased the size of paw edema at 1, 3, and 5 h, producing 71, 54, and 43% inhibition of paw swelling when compared with the DI water control group. MPE at doses of 250, 500, and 1000 mg/kg BW reduced the volume of paw swelling; however, only a dose of 1000 mg/kg BW presented a significant effect at all evaluation times. MPE at a dose of 1000 mg/kg BW demonstrated 52, 44, and 30% inhibition of paw edema at 1, 3, and 5 h, respectively. These findings suggest that MPE exhibited anti-inflammatory properties through the suppression of paw edema in rats.

**Table 5.** Anti-inflammatory effects of MPE on carrageenan-induced hind paw edema in rats. Data are expressed as mean $\pm$ SD values (n = 6). Accordingly, * $p < 0.05$ when compared with DI water.

| Group | Measure | Time (h) | | |
|---|---|---|---|---|
| | | **1** | **3** | **5** |
| DI water (5 mL/kg BW) | Volume (mm³) | 306.0 ± 29.4 | 388.7 ± 23.8 | 473.3 ± 16.3 |
| Aspirin (300 mg/kg BW) | Volume (mm³) Inhibition (%) | 92.7 ± 8.5 * 71 | 180.7 ± 21.9 * 54 | 272.7 ± 35.6 * 43 |
| MPE (250 mg/kg BW) | Volume (mm³) Inhibition (%) | 299.3 ± 33.1 3 | 367.3 ± 31.2 5 | 357.3 ± 31.0 23 |
| MPE (500 mg/kg BW) | Volume (mm³) Inhibition (%) | 250.0 ± 15.3 19 | 357.3 ± 31.0 8 | 404.7 ± 28.3 15 |
| MPE (1000 mg/kg BW) | Volume (mm³) Inhibition (%) | 149.3 ± 11.0 * 52 | 224.0 ± 23.1 * 44 | 334.7 ± 19.4 * 30 |

Abbreviations: DI = deionized, MPE = Mahajanaka mango pulp extract.

### 3.5. Protective Effects of MPE against Gastric Ulcers in Rats

Water immersion and restraint stress (WIRS) led to gastric ulcers in rats. Apparently, a reference dosage of NSAID cimetidine (100 mg/kg BW) potently diminished the length of the gastric ulcer, while MPE (250, 500, and 1000 mg/kg BW) also did so independently of the dose, indicating inhibitory values of the gastric ulceration at 92.5, 10.9, 20.0, and 19.7%, respectively, when compared with the DI water treatment (Table 6 and Figure 2A).

**Table 6.** Gastroprotective effects of MPE on WIRS-, HCl/ethanol-, or indomethacin-induced rats. Data are expressed as mean $\pm$ SD values (n = 6). Accordingly, * $p < 0.05$ when compared with DI water.

| Group | Measure | Gastric Ulcer Induction | | |
|---|---|---|---|---|
| | | **WIRS** | **HCl/Ethanol** | **Indomethacin** |
| DI water (5 mL/kg BW) | Ulcer length (mm) | $11.82 \pm 1.43$ | $78.00 \pm 13.30$ | $12.97 \pm 5.42$ |
| Cimetidine (100 mg/kg BW) | Ulcer length (mm) Inhibition (%) | $0.88 \pm 0.38$ * 92.5 | $18.50 \pm 9.74$ * 76.3 | $3.60 \pm 0.87$ * 72.2 |
| MPE (250 mg/kg BW) | Ulcer length (mm) Inhibition (%) | $10.53 \pm 2.24$ 10.9 | $44.17 \pm 14.15$ 43.4 | $9.97 \pm 2.52$ 23.1 |
| MPE (500 mg/kg BW) | Ulcer length (mm) Inhibition (%) | $9.45 \pm 2.04$ 20.0 | $41.17 \pm 16.67$ 47.2 | $9.60 \pm 1.04$ 26.0 |
| MPE (1000 mg/kg BW) | Ulcer length (mm) Inhibition (%) | $9.48 \pm 2.20$ 19.7 | $47.83 \pm 13.98$ 38.7 | $8.92 \pm 1.12$ 31.2 |

Abbreviations: DI = deionized, HCl = hydrochloric acid, MPE = Mahajanaka mango pulp extract, WIRS = water immersion and restraint stress.

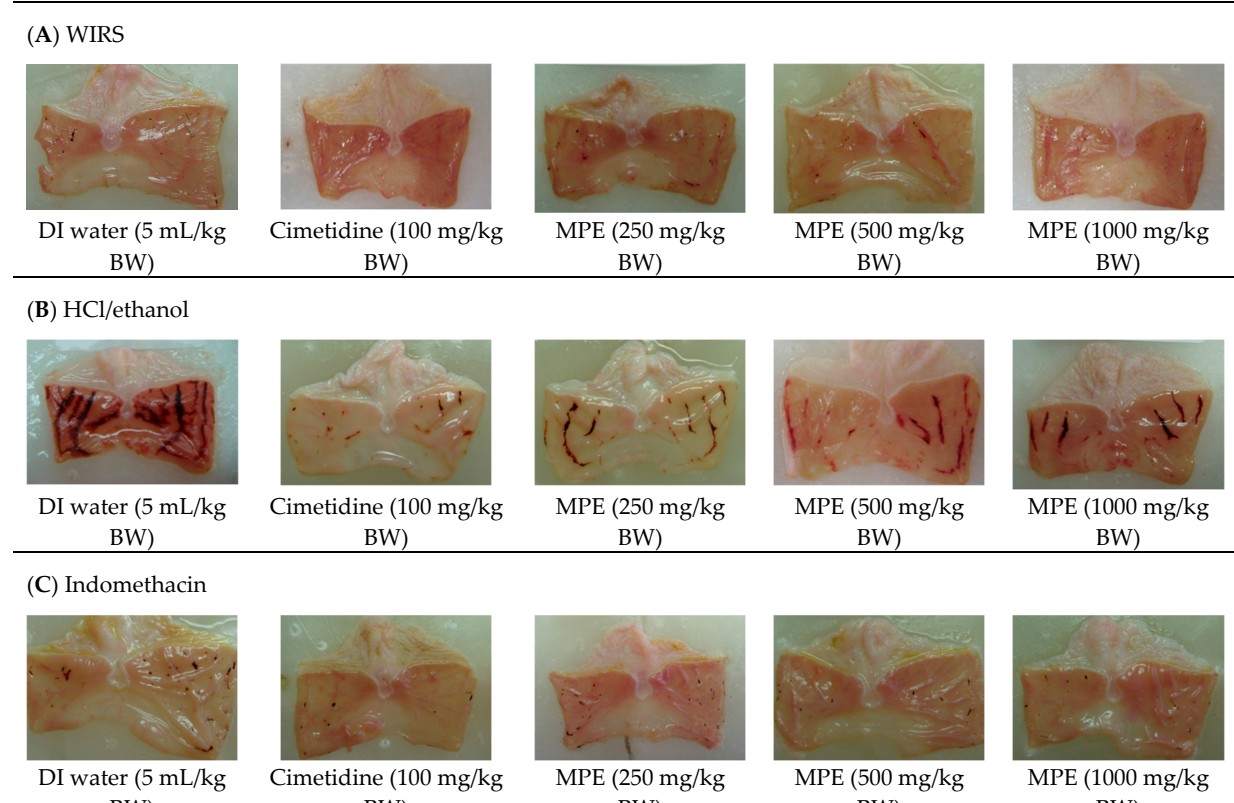

(**A**) WIRS

| DI water (5 mL/kg BW) | Cimetidine (100 mg/kg BW) | MPE (250 mg/kg BW) | MPE (500 mg/kg BW) | MPE (1000 mg/kg BW) |

(**B**) HCl/ethanol

| DI water (5 mL/kg BW) | Cimetidine (100 mg/kg BW) | MPE (250 mg/kg BW) | MPE (500 mg/kg BW) | MPE (1000 mg/kg BW) |

(**C**) Indomethacin

| DI water (5 mL/kg BW) | Cimetidine (100 mg/kg BW) | MPE (250 mg/kg BW) | MPE (500 mg/kg BW) | MPE (1000 mg/kg BW) |

**Figure 2.** Macroscopic appearance and ulcerative area of gastric mucosa from rats induced by WIRS (**A**), HCl/ethanol (**B**), and indomethacin (**C**). Abbreviations: DI water = deionized water, HCl = hydrochloric acid, MPE = Mahajanaka mango pulp extract.

In addition, administration of HCl/ethanol caused severe gastric ulcers in rats, while cimetidine significantly inhibited gastric ulceration at 76.3%. The consumption of MPE (250, 500, and 1000 mg/kg BW) inhibited HCl/ethanol-induced ulceration at 43.4, 47.2, and 38.7%, respectively, when compared with the DI water treatment (Table 6 and Figure 2B). Likewise, the ingestion of the NSAID indomethacin caused gastric ulcers, whereas cimetidine showed a 72.2% inhibition of gastric ulceration. Moreover, MPE (250, 500, and 1000 mg/kg BW) diminished indomethacin-induced gastric ulcers in a concentration-dependent manner, indicating 23.1, 26.0, and 31.2% inhibition, respectively, when compared with the DI water treatment (Table 6 and Figure 2C). Taken together, these findings imply that MPE consumption had gastroprotective effects on gastric ulceration in rats induced by stress, over-secretions of hydrochloric acid and ethanol, and consumption of high doses of an NSAID.

### 3.6. Hepatoprotection of MPE in Chemical-Induced Rats

With regard to liver enzymes (Table 7), including alanine aminotransferase (ALT) and aspartate aminotransferase (AST), the consumption of MPE (1000 mg/kg BW) was found to slightly decrease serum levels of ALT and AST activities but non-significantly when compared with DI water treatment (no induction group). This suggests that the high dose of MPE exhibits no hepatotoxicity in the rats. The injection of $CCl_4$ (*ip*) increased levels of ALT considerably and that of AST slightly when compared with subjects that had not received the induction with $CCl_4$ (*ip*). Silymarin, which is a flavonoid extracted from milk thistle seeds, was used as a standardized antioxidant. Accordingly, treatments of silymarin (100 mg/kg BW, *po*) restored an increased level of ALT markedly and that of AST slightly, while the consumption of MPE (250, 500, and 1000 mg/kg BW) restored the increased levels of ALT tentatively in a concentration-dependent manner but did not influence the levels of AST when compared with the DI water control treatment. In addition, the administration of a high dose of paracetamol significantly increased the levels of ALT and AST when compared with DI water treatment (no induction group). Surprisingly, MPE (250, 500, and 1000 mg/kg BW), as well as silymarin (100 mg/kg BW, *po*), restored the increased serum levels of ALT and AST independently of the dosages; however, MPE was found to be more effective than silymarin.

**Table 7.** Serum levels of AST and ALT activity in $CCl_4$- or paracetamol-induced hepatotoxicity in rats. Data are expressed as mean $\pm$ SD values (n = 8). Accordingly, * $p < 0.05$ when compared without the induction; # $p < 0.05$ when compared with the hepatotoxicity induction.

| Induction | Treatment | AST (IU/L) | ALT (IU/L) |
|---|---|---|---|
| No | DI water (5 mL/kg BW) | 87 $\pm$ 40 | 37 $\pm$ 33 |
|  | MPE (1000 mg/kg BW) | 62 $\pm$ 15 | 24 $\pm$ 17 |
| $CCl_4$ (2 mg/kg BW) | DI water (5 mL/kg BW) | 99 $\pm$ 30 | 126 $\pm$ 122 |
|  | Silymarin (100 mg/kg BW) | 87 $\pm$ 48 | 54 $\pm$ 31 |
|  | MPE (250 mg/kg BW) | 98 $\pm$ 52 | 62 $\pm$ 26 |
|  | MPE (500 mg/kg BW) | 117 $\pm$ 84 | 60 $\pm$ 62 |
|  | MPE (1000 mg/kg BW) | 119 $\pm$ 33 | 30 $\pm$ 25 |
| Paracetamol (2.5 g/kg BW) | DI water (5 mL/kg BW) | 278 $\pm$ 118 * | 553 $\pm$ 219 * |
|  | Silymarin (100 mg/kg BW) | 281 $\pm$ 161 | 417 $\pm$ 297 |
|  | MPE (250 mg/kg BW) | 212 $\pm$ 91 | 200 $\pm$ 110 # |
|  | MPE (500 mg/kg BW) | 266 $\pm$ 95 | 374 $\pm$ 188 |
|  | MPE (1000 mg/kg BW) | 234 $\pm$ 138 | 247 $\pm$ 222 |

Abbreviations: $CCl_4$ = carbon tetrachloride, DI = deionized, MPE = Mahajanaka mango pulp extract.

Liver wet weights of all rat groups showed no significant differences (Table 8). According to histochemical hematoxylin and eosin (H&E) staining, treatment with the highest dose of MPE (1000 mg/kg BW) did not result in morphological changes in the liver tissue (B) when compared with DI water treatment (A). Obviously, $CCl_4$ induction (*ip*)/DI water

treatment (*po*) caused hepatoxicity, which was indicated by a severe degree of fatty changes and fatty cysts in the liver tissue (C), whereas pretreatments of silymarin (100 mg/kg BW, *po*) effectively decreased the degree of fatty changes and fatty cysts in the liver when compared with the CCl$_4$/DI water group. Indeed, MPE (250 and 1000 mg/kg BW, *po*) lowered the degree of lipid accumulation in the liver; inversely, MPE (500 mg/kg BW, *po*) exhibited a severe degree of lipid accumulation. Apparently, taking paracetamol (2 g/kg BW) induced a severe degree of hepatic necrosis when compared with rats that had not received the treatment. Similarly, pretreatments of silymarin (100 mg/kg BW, *po*) and MPE (250 and 1000 mg/kg BW, *po*) revealed a mild degree of hepatic necrosis, while MPE (500 mg/kg BW, *po*) revealed a moderate to severe degree of hepatic necrosis. Accordingly, the findings imply that the consumption of MPE (1000 mg/kg BW) was not harmful to the liver of healthy rats, while MPE (250 mg/kg BW) did improve fat deposition in the livers of CCl$_4$-fed rats and offered protection against liver cell damage in paracetamol-fed rats.

**Table 8.** Wet weight values (mean ± SD) and H&E staining of liver from rats with hepatotoxicity induced by CCl$_4$ and paracetamol.

| Induction | Code | Treatment | Wet Weight (g/100 g BW) | H&E Staining |
|---|---|---|---|---|
| No | A | DI water (5 mL/kg BW) | ND |  |
| | B | MPE (1000 mg/kg BW) | ND |  |
| CCl$_4$ (2 mg/kg BW) | C | DI water (5 mL/kg BW) | 4.93 ± 0.18 |  |
| | D | Silymarin (100 mg/kg BW) | 4.83 ± 0.15 |  |
| | E | MPE (250 mg/kg BW) | 4.83 ± 0.09 |  |
| | F | MPE (500 mg/kg BW) | 4.97 ± 0.20 |  |
| | G | MPE (1000 mg/kg BW) | 4.78 ± 0.22 |  |

**Table 8.** *Cont.*

| Induction | Code | Treatment | Wet Weight (g/100 g BW) | H&E Staining |
|---|---|---|---|---|
| Paracetamol (2.5 g/kg BW) | H | DI water (5 mL/kg BW) | $4.20 \pm 0.13$ |  |
| | I | Silymarin (100 mg/kg BW) | $3.89 \pm 0.14$ |  |
| | J | MPE (250 mg/kg/kg BW) | $4.05 \pm 0.09$ |  |
| | K | MPE (500 mg/kg BW) | $4.05 \pm 0.09$ |  |
| | L | MPE (1000 mg/kg BW) | $4.11 \pm 0.13$ |  |

Abbreviations: BW = body weight, $CCl_4$ = carbon tetrachloride, MPE = Mahajanaka mango pulp extract, ND = not determined.

## 4. Discussion

Previous studies have investigated the chemical and biological properties of different parts of the mango, including the pulp, peel, leaves, bark, and stem [4,31,32]. Mangoes are known to contain various bioactive compounds such as mangiferin, gallotannins, gallic acid, quercetin, and catechins, which are present in different amounts depending upon the cultivar, the part of the mango, its ripeness, and the extraction method employed [3,4,33]. The present study focused on extracting ripe Mahajanaka mango pulp using distilled water, and it was found that the MPE contained a richness of mangiferin, β-carotene, and vitamin C, along with phenolic compounds, particularly quercetin, kaempferol, and their derivatives. These findings correspond with the compounds that have been previously reported to be present in other mango varieties [32].

Pain serves as an essential protective mechanism in the human body. Pain signals are generated in response to noxious stimuli such as physical injuries, inflammation, or tissue damage [34]. The transmission of these signals to the central nervous system involves the activation of nociceptors, which are specialized sensory neurons that are capable of detecting and responding to harmful stimuli [35]. Upon activation, nociceptors release a range of chemical mediators, including prostaglandins, bradykinin, and histamine, which contribute to the perception of pain [36]. In this research study, the analgesic properties of MPE were evaluated using the acetic acid-induced writhing test. The results showed a significant dose-dependent reduction in the number of writhes, suggesting that MPE has the potential to alleviate pain. This effect may be attributed to the presence of phytochemicals in MPE that can inhibit nociception and consequently reduce acute pain. Previous studies have reported the peripheral antinociceptive activity of isolated mangiferin derived from mango bark, or the ethanolic extract of mango leaves, against acetic acid-induced nociception in mice [37,38]. Another study revealed that mangiferin exhibited a

robust ability to suppress the activation of microglia induced by LPS, which was mediated through the downregulation of cyclooxygenase 2 (COX-2) and the subsequent synthesis of prostaglandin E2 [39]. Indeed, β-carotene has been shown to possess analgesic activity by reducing the number of acetic acid-induced writhing responses in mice [40]. These findings suggest that the phytochemicals in MPE, including mangiferin and β-carotene, may contribute to its analgesic effects.

In addition to its potential analgesic properties, it is important to note that the mango extracts may also have anti-inflammatory effects. Inflammation can lead to the development of edema, which is characterized by an abnormal accumulation of fluid in the interstitial spaces of injured tissues [41]. This process is mediated by the release of inflammatory mediators, such as histamine and prostaglandins, which cause vasodilation and increased vascular permeability [42,43]. This allows fluid and immune cells to escape from blood vessels and accumulate at the site of the injury. Furthermore, inflammatory cells, such as neutrophils and macrophages, can amplify the development of edema through the release of cytokines like interleukin (IL)-1β, IL-6, and tumor necrosis factor-alpha (TNF-α) [10,41]. The present study investigated the potential anti-inflammatory effects of MPE using two established rat models, namely EPP-induced ear edema and carrageenan-induced hind paw edema. Our results demonstrate that MPE treatment significantly attenuated both types of edemas in rats, suggesting a potential anti-inflammatory activity of the mango extract. These findings are consistent with previous studies that have shown that topical applications of mango leaf extract containing 31% mangiferin significantly decreased ear edema induced by EPP in rats [44]. Based on a study conducted on mice, mango stem bark extract was found to be effective in reducing ear edema caused by arachidonic acid (AA) and phorbol myristate acetate (PMA) through inhibition of TNF-α levels and myeloperoxidase (MPO) activity [45]. Additionally, the mango stem bark extract, along with mangiferin, was also found to suppress the levels of prostaglandin E2 and leukotriene B4 in an LPS-stimulated macrophage cell line (RAW264.7). The outcomes of these studies provide further evidence of the potential anti-inflammatory effects of MPE containing mangiferin, which may be attributed to the suppression of inflammatory mediators and proinflammatory cytokines.

Peptic ulcers are caused by an imbalance between the protective factors of the gastric mucosa and the harmful effects of gastric acid and pepsin [46]. The development of peptic ulcers is a complex process influenced by various factors, including Helicobacter pylori infection, the use of NSAIDs such as ibuprofen and indomethacin, excessive alcohol consumption, smoking, and stress [47,48]. These factors can cause erosion of the gastric mucosal barrier, leading to inflammation and ulcer formation [49]. We evaluated the gastroprotective effects of MPE using water immersion restraint stress-, HCl/ethanol-, and indomethacin-induced gastric ulcers in rats. Our findings suggest that MPE exhibited a greater tendency to lower gastric ulcer formation in the HCl/ethanol model than in the indomethacin and WIRS models. It is important to note that each stimulator, such as stress, alcohol, or certain drugs, can promote the development of gastric ulcers through a distinct mechanism in the body, while specific medications may inhibit a particular marker involved in the development of gastric ulcers. Although the gastroprotective effects of MPE were not as potent as those of cimetidine (positive control) in protecting against gastric ulcers, it should be considered that cimetidine could serve as a specifically designed medication for reducing acid secretion in the stomach by blocking the histamine H2 receptor [50,51]. In contrast, MPE contains a mixture of different compounds that may produce varying gastroprotective effects. A previous study found that β-carotene exhibited a protective effect against indomethacin-induced gastric ulcers in rats through decreasing gastric mucosal lesions and the restoration of the gastric mucosal redox state [52]. Indeed, several studies have demonstrated that mango extract, or mangiferin, exhibited potent anti-inflammatory and antioxidant properties, which may effectively reduce inflammation and oxidative stress in the gastric mucosa [4,44,53]. As these factors are implicated in the development of gastric ulcers, MPE may facilitate wound healing in peptic ulcers by

reducing them. According to the findings of several studies, mangiferin has been shown to facilitate wound closure in excision or incision wounds through the downregulation of COX-2, as well as by stimulating angiogenesis by increasing the secretion of the vascular endothelial growth factor (VEGF) [54–56]. This suggests that MPE may have a multifaceted mechanism of action, making it a potentially useful treatment option for promoting the healing of peptic ulcers. However, it is essential to note that the anti-inflammatory activity of MPE, like other NSAIDs, may not be suitable for use in the early stages of peptic ulcer occurrence since it may inhibit the function of prostaglandins, which can help to suppress HCl secretions in the stomach. Further studies should be conducted to investigate the underlying mechanisms of MPE for the treatment of peptic ulcers at higher doses.

Liver injury refers to the impairment of liver function and structure, which can be caused by a variety of factors including viral infections and alcohol consumption, and can be attributed to the use of certain medications such as $CCl_4$ and paracetamol [57,58]. In the present study, we investigated the effect of MPE on $CCl_4$- and paracetamol-induced liver injuries in rat models. $CCl_4$ is metabolized by the liver into a highly reactive free radical called trichloromethyl radical ($\bullet CCl_3$), while paracetamol, which is also known as acetaminophen, undergoes metabolism via the cytochrome P450 pathway leading to the production of N-acetyl-p-benzoquinone imine (NAPQI), a highly reactive and toxic metabolite [59,60]. Both of these can cause oxidative stress, inflammation, and lipid accumulation in the liver, ultimately resulting in hepatic steatosis and liver damage [61]. Indeed, liver damage causes a leakage of liver enzymes including AST, ALT, and alkaline phosphatase (ALP). Our present study demonstrated that the administration of MPE at 250 mg/kg BW effectively reduced levels of ALT and fat deposits in rats with both $CCl_4$- and paracetamol-induced hepatotoxicity. It is likely that the hepatoprotective effects of MPE containing mangiferin may be attributed to its ability to scavenge lipid radicals and consequently reduce fat deposits. This hypothesis is supported by the outcomes of previous studies that reported the hepatoprotective effects of mangiferin isolated from mango bark by reducing the levels of ALP, ALT, triglycerides, total cholesterol, and lipid peroxidation while increasing reduced glutathione (GSH) in rats with galactosamine-induced liver damage [62]. Another previous study found that the administration of mangiferin also improved LPS- and D-galactosamine-induced acute liver injuries in mice via suppression of the levels of serum ALT, AST, IL-1β, and TNF-α, as well as hepatic malondialdehyde (MDA) and reactive oxygen species (ROS) levels [63]. Additionally, mangiferin potently reduced fat accumulation, as well as the number of Kupffer cells and the expression of nuclear factor kappa B (NF-κB), in the livers and adipose tissues of high-fat diet (HFD)-induced metabolic mice [64]. Furthermore, Morakinyo et al. demonstrated that β-carotene significantly improved paracetamol-induced liver damage in rats by reducing the levels of AST, while increasing superoxide dismutase (SOD), catalase, and GSH activities [65]. Another study found that vitamin C also protected against paracetamol-induced hepatotoxicity in rats by ameliorating the levels of ALT, AST, and lipid peroxidation [66]. These data are consistent with the antioxidant and anti-inflammatory activities of mangiferin, β-carotene, and vitamin C in MPE. These findings suggest that MPE, containing mangiferin, β-carotene, and vitamin C, has the potential to protect the liver against $CCl_4$- and paracetamol-induced liver injuries by reducing liver enzymes and fat deposits.

Taken together, these findings indicate that Mahajanaka mango, containing mangiferin, β-carotene, and vitamin C, possesses a positive protective effect against oxidative stress, pain, inflammation, gastric ulceration, and hepatotoxicity. The results of this study provide supportive evidence for the antioxidant, analgesic, anti-inflammatory, gastroprotective, and hepatoprotective properties of Mahajanaka mango, which could be an effective agent against oxidative stress and inflammation-associated diseases. Further studies are warranted to uncover the underlying mechanisms and evaluate its clinical efficacy against parameters of oxidative stress and inflammation-associated diseases.

## 5. Conclusions

A water extract derived from Mahajanaka mango (*M. indica*) pulp, rich in mangiferin, β-carotene, and vitamin C, could effectively reduce acetic acid-induced writhing responses in mice; inhibit EPP-induced ear edema and carrageenan-induced hind paw edema in rats; prevent gastric ulceration induced by stress, hydrochloric acid/ethanol and indomethacin in rats; and prevent lipid accumulation and inflammation in the livers of $CCl_4$- and paracetamol-induced hepatotoxic rats. The pulp extract exerted analgesic, anti-inflammatory, gastroprotective, and hepatoprotective effects in murine models while providing a number of health benefits. However, further research will be necessary to fully evaluate the fruit extract's biological and pharmacological properties in humans.

**Author Contributions:** Conceptualization, S.S. and P.K.; methodology, D.K.; validation, D.K. and T.T.; formal analysis, K.S.; investigation, R.J.; data curation, T.T.; writing—original draft preparation, N.P.; writing—review and editing, N.P. and P.K.; visualization, P.K.; supervision, D.K.; project administration, S.S.; funding acquisition, S.S. All authors have read and agreed to the published version of the manuscript.

**Funding:** This work was funded by the Thailand Research Fund (Agriculture Section), Grant Number RDG5420069.

**Institutional Review Board Statement:** The animal study protocol was approved by the animal ethical committee of the Medical Faculty, Chiang Mai University, Thailand (Protocol Number 21/2554).

**Informed Consent Statement:** Not applicable.

**Data Availability Statement:** All relevant data are included within the manuscript.

**Acknowledgments:** We acknowledge the Ethical Committee for Animal Study from the Faculty of Medicine, Chiang Mai University, for approving the study protocol and the Animal Housing Unit of the Medical Faculty for generously granting us access to their facilities.

**Conflicts of Interest:** The authors declare no conflict of interest.

## Abbreviations

| | |
|---|---|
| AA | arachidonic acid |
| ALT | alanine aminotransferase |
| ALP | alkaline phosphatase |
| AST | aspartate aminotransferase |
| $CCl_4$ | carbon tetrachloride |
| COX-2 | cyclooxygenase-2 |
| DI | deionized |
| DMSO | dimethyl sulfoxide |
| EPP | ethyl phenylpropiolate |
| GAE | gallic acid equivalent |
| GSH | glutathione |
| H&E | hematoxylin and eosin |
| HFD | high-fat diet |
| h | hour |
| HCl | hydrochloric acid |
| ip | intraperitoneal |
| LPS | lipopolysaccharide |
| IL | interleukin |
| MDA | malondialdehyde |
| MPE | Mahajanaka mango pulp extract |
| MPO | myeloperoxidase |
| NAPQI | n-acetyl-p-benzoquinone imine |
| NSAIDs | nonsteroidal anti-inflammatory drugs |
| NF-κB | nuclear factor kappa light chain enhancer of activated B cells |
| PMA | phorbol myristate acetate |

| po | per oral |
|---|---|
| ROS | reactive oxygen species |
| sc | subcutaneously |
| SD | standard deviation |
| $\bullet CCl_3$ | trichloromethyl radical |
| TEAC | Trolox-equivalent antioxidant capacity |
| TNF-$\alpha$ | tumor necrosis factor-alpha |
| VEGF | vascular endothelial growth factor |
| WIRS | water immersion and restraint stress |

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
