# Peer review of "Antioxidant and Biological Activities of Mahajanaka Mango Pulp Extract in Murine Models"

_applsci, doi:10.3390/app132212216_

Round 1

Reviewer 1 Report

Comments and Suggestions for Authors

The article generally presents originality, its methodological design is adequate and its results are significant.

Author Response

Please find an attached file

Reviewer 2 Report

Comments and Suggestions for Authors

1. The citation is started from 8. Please revise the citation serial throughout the manuscript (Line 31).

2. Include accession number of plant/ fruit from concerned authority/ taxonomist. Without proper accession number of the claim is not so strong. 

3. Include " DPPH/ Free radical scavenging activity testing"  (reference: https://doi.org/10.1186/s12906-021-03290-6) to strengthen antioxidant activity.

4. Cytotoxicity activity is highly related to antioxidant activity. Please add this activity too (reference: https://doi.org/10.1016/j.jsps.2020.11.004 ) 

5. Please add a future recommendations section before conclusion.

6. Few minor typos are still in the manuscript. Please check the issue carefully.

7. Abbreviations should be represented in two column system. It will be easier to check then.

Comments on the Quality of English Language

Minor correction is needed. 

Author Response

Please find an attached file

Reviewer 3 Report

Comments and Suggestions for Authors

The subject of the article was the analysis of the antioxidant and biological activities of Mahajanaka mango pulp extract. The text is written in correct English, the structure is simple and clear. The measurement was carried out correctly, all the necessary parameters were determined and checked.

I recommend accepting the article after minor revision. I am asking the authors to respond and make appropriate changes to the text, to the following comments:

1) Introduction – line 33 and following – citing numbering does not appear in order. They start with number eight, and number one doesn't appear until the next paragraph.

2) The chromatogram (Fig. 1) lacks the assignment of signals to individual components (Table 1 contains only six compounds).

3) Table 7 - How do the authors explain such high measurement uncertainty?

4) Why was the content of other ingredients such as quercetin or kaempferol not determined? GAE is not an accurate measure of polyphenol content.

5) A very valuable tool for analyzing the results of many studies is principal component analysis (PCA). It would, for example, allow the authors to answer the question which of the components is responsible for specific biological properties. The use of this tool would certainly increase the value of the publication.

Author Response

Please find an attached file
